# Social Prestige of the Paramedic Profession

**DOI:** 10.3390/ijerph18041506

**Published:** 2021-02-05

**Authors:** Anita Majchrowska, Jakub Pawlikowski, Mariusz Jojczuk, Adam Nogalski, Renata Bogusz, Luiza Nowakowska, Michał Wiechetek

**Affiliations:** 1Department of Humanities and Social Medicine, Medical University of Lublin, 7 Chodźki Str., 20-093 Lublin, Poland; jakub.pawlikowski@umlub.pl (J.P.); renata.bogusz@umlub.pl (R.B.); luiza.nowakowska@umlub.pl (L.N.); 2Medical Faculty, University of Cardinal Stefan Wyszynski in Warsaw, 5 Dewajtis Str., 01-815 Warsaw, Poland; 3Department of Trauma Surgery and Emergency Medicine, Medical University of Lublin, 11 Staszica Str., 20-081 Lublin, Poland; mariusz.jojczuk@umlub.pl (M.J.); adam.nogalski@umlub.pl (A.N.); 4Faculty of Psychology, The John Paul II Catholic University of Lublin, 14 Racławickie Av., 20-950 Lublin, Poland; wiechetek@kul.lublin.pl

**Keywords:** paramedics, social prestige, respect for profession

## Abstract

Background: There is a lack of research on social image, prestige, and the position of the paramedic profession in the social structure. The main objective of the study was to determine the place of the paramedic profession in the hierarchy of prestige of professions as viewed by the public. In operationalizing the term ‘prestige’, we deemed the word ‘respect’ to best fit the sense of the subjective evaluation of prestige with regard to a profession. Material and methods: The data comes from cross-sectional survey-based research. The research was carried out on a group of 600 people over 18 years of age. The sample was of a random nature, and the selection of respondents was calculated on the basis of them being representative of the Polish population. Results: The median of respect declared for the paramedic profession, on a scale of 1 to 5, was 4.49, which placed the profession in fourth place in the ranking. The assessment of respect for paramedics among other medical professions placed them in third place, directly after doctors and midwifes. Conclusions: The profession of paramedic is characterized by high social prestige, locating it at the forefront of the medical profession and other examined professions, but its social position, expressed by objective measures (earnings, structural possibilities, social power), is significantly lower.

## 1. Introduction

Paramedics, along with doctors and nurses, form the core of the modern emergency medical system. The profession of paramedic was finally regulated in Poland, in 2006, by the Emergency Medical Care Act [1] and the Regulations of the Minister of Health, setting the scope of professional activities performed by paramedics employed in the emergency medical care system [2]. Although the profession itself is a new one, subsequent legal acts have increased the autonomy and professional independence of paramedics, for example, in the scope of drugs administered, as well as the scope of medical activities that are allowed to be undertaken. Currently, a Polish paramedic can independently perform 28 types of emergency medical procedures, provide 29 different healthcare services other than emergency medical procedures, and administer 47 types of drugs [2].

In other European countries, for example in the UK, the direction of change is similar, with a gradual increase in the scope of professional independence [3,4]. The demand for care of paramedics is constantly growing, albeit the percentage of needs directly related to life-threatening situations is falling. Only 10% of emergency calls are seen to be life-threatening in the UK [5]. The direction of these changes redefines the role of a paramedic, broadening its scope from direct lifesaving procedures towards broadly understood patient care, whether pre-hospital, in the patient’s environment, or in health promotion [5,6,7,8].

Even though the profession of paramedic has a regulated legal status and scope of competencies, it is still on its path to full professionalization and professional identity [3,4,5,6,7]. According to the classic concept of Eliot Freidson, professions have a dominant position in the labor market, are independent in the performance of professional activities, and have high ethical standards and professional qualifications. These features provide the profession with great respect and high social position [9]. The progressive professionalization of paramedic in Poland finds its expression, among other things, in the growing requirements for qualifications of those who perform it. Currently, one needs a higher education diploma with a bachelor’s or master’s degree or needs to pass the State Paramedic Examination. Paramedics are also obliged by law to improve their professional skills and to systematically improve their qualifications. 

Despite the above, the process of professionalization of the paramedic profession seems to be incomplete. This is indicated by the lack of professional self-government or an act on the profession and the code of ethics to regulate the functioning in a profession where the ethical-moral requirements are indivisible from the essence of professional activities. However, legal and organizational changes tend to make them well-educated and competent members of the medical staff, and their scope of duties and professional independence has been systematically increasing [10,11,12,13,14,15].

Both in Poland and in Europe, there is a lack of research on social image, prestige, and the position of the paramedic profession in the social structure. Most often, sociological analyses of the profession of paramedic concentrate on organizational culture and the place of paramedics within the healthcare system [16,17,18,19], the professionalization of this profession [20,21], methods and conditions for performing this professional role [22,23,24,25,26,27,28,29], qualifications [5,30] and the psychosocial aspects related to working as a paramedic [31,32,33,34,35,36,37]. 

Paramedics rarely appear on ranking scales in the research on the prestige of professions, which, in addition to socio-economic status, is an important measure of the position of the profession in the stratification of the respective society [38]. The social perception of the profession of paramedic, its prestige, and its social position determines a patient’s attitude to the paramedic as a representative of the health care system and a member of the interdisciplinary team working for the health of the patient. 

The prestige of a profession as a sociological category is formed from a subjective assessment, including the elements of esteem, respect, and dignity [39]. The diversity of prestige forms a kind of a ‘stratification ladder’, which is a manifestation of social stratification in the sphere of the collective consciousness. The assessment of the prestige of a profession consists of various subjective factors, which include an awareness of the benefits, rewards, and values that are attributed to the respective professions [40]. Although prestige is primarily a subjective category, it can also reflect the objective dimensions of social positioning, based on the level of income and education and broadly understood structural possibilities. According to Meyer and Hammond, a social position or social status has a variety of stratified properties or status attributes. The most commonly used attributes are: allocated social resources-authority (power, funds, contacts, and other rights needed to perform activities); reward-pay in various forms, and other gratifications; cultural valuation of a status and its activities (respect, prestige, or standing, which attach variously to different statuses); market value of, or scarcity of, status activities-value, as established in economic and other social markets); required features or investments of status occupants—the amount of education, skills, time, etc.) [41].

In a 1979 Polish study on the prestige of professions, paramedics were grouped with nurses and midwives and scored an average of 46.0 (Normalized average grade with 100 as the highest value and 0 as the lowest). In the same ranking, doctors scored 78.0. In 2009, the same group of professions obtained a higher average grade, reaching the level of 55.9 (doctors 87.8) [42]. In turn, in a 1989 American study, using the same scale, paramedics obtained an average score on the ranking list of 61.2, while the average score for doctors was 86.05 [43]. 

Ina 2012 study, paramedics scored 6 points on a 1 to 9 scale (doctors 7.6) [44]. According to Forbes, in the 2016 assessment of the most prestigious American professions, paramedics occupied a relatively high position, eighth, following doctors, scientists, firemen, army officers, engineers, nurses, and architects [45]. 

Patient satisfaction surveys conducted among those who required the assistance of paramedics in various health situations indicate a high assessment of their attitude and competence. Finnish research demonstrates that the level of satisfaction with contact with paramedics was very high and evaluated, among others, dimensions such as: communication with the patient, diagnosis of patient condition, procedures applied, satisfying information needs, and the general behavior of paramedics [46]. In American research by Crowe et al., most people using the services of paramedics rated them as ‘excellent’ [47], while in the research by Bernard et al., almost all patients were either satisfied or very satisfied with their contact with the emergency medical team [48]. Similar results were obtained in England [49]. The aforesaid studies, however, lack the features of representativeness, as they apply different methodologies and scoring schemes and hence they can only signal the social attitudes of selected groups towards the paramedics.

The main objective of the present research was to determine the position of the paramedic profession in the prestige hierarchy of selected professions by analyzing the levels of its social respect and regard in public opinion. An additional objective of the study was to assess the paramedic’s income and to indicate the relationship between the assessment of the social prestige of a paramedic and selected sociodemographic characteristics of the respondents. The research was carried out on a sample that was representative for the whole country. 

## 2. Material and Methods

### 2.1. Study Participants

The research was carried out on a group of 600 people over 18 years of age. The sample was of a random nature and the selection of respondents was calculated on the basis of them being representative of the Polish population in the following areas: sex of respondents (100% compliance with calculations based on the Local Data Bank (LDB)), age of respondents (maximum deviation of 2% from calculations based on the LDB), the number of respondents in a given voivodship calculated on the basis of the population distribution throughout the country (100% compliance with calculations based on the LDB), place of residence (maximum deviation of 1% from calculations based on the LDB), and level of education (maximum deviation of 3% from calculations based on the LDB).

The sample was primarily selected using random-route as a default method (employing the computer-assisted personal interview (CAPI) technique), while gaps in the metrical data (collected using the computer-assisted telephone interviewing (CATI) technique) were filled in by using a number generator to draw telephone numbers from a database of active numbers issued by Polish landline and mobile operators. The maximum acceptable statistical error of measurement was 4%, with a confidence interval of 95%. 

The structure of the studied group in terms of sociodemographic features was consistent with the structure of the Polish population and held a risk of statistical error of 4%. The detailed characteristics of the studied group are presented in Table 1.

### 2.2. Data Collection

The data are derived from cross-sectional survey-based research carried out in 2018, using a mixed-mode survey technique comprising 84% CAPI (the default technique) and 16% CATI (a secondary technique used to supplement the established metrical distributions). The study was a part of a wider research project on the perception of medical professions in Poland. 

In operationalizing the term ‘prestige’, translating it into the language of research questions, we deemed the word ‘respect’ to best fit the sense of the subjective evaluation of prestige with regard to a profession. We patterned this on the periodic research on prestige of professions that were conducted in Poland by the Public Opinion Research Centre. In the description of research results, the terms ‘prestige’ and ‘respect’ are used interchangeably.

Respondents were asked what level of respect they had for the representatives of twenty different professions. For each of the professions, the respondents defined the level of their respect on a 1–5 scale, as follows: 1—very low respect, 2—small, 3—average, 4—large, 5—very high respect.

The same scale was used in the question: “What degree of respect do you have for the following medical professions?” Here, 12 widely recognized medical professions were selected for assessment: doctor, dentist, nurse, midwife, pharmacist, laboratory diagnostician, paramedic, feldsher, physiotherapist, dietitian, cosmetologist, and psychotherapist.

Respondents were also asked to rate the paramedics income on a scale of 1 to 5, as follows: 1—very low earnings, 2—low, 3—average, 4—high, 5—very high earnings. 

### 2.3. Statistical Analysis

Analysis was carried out using frequency measurements and percentage and median indicators (with an option in which values are group midpoints) for independent variables as well as the respondent’s answers. The role of independent variables was assigned to sex, age, level of education, place of residence, self-assessment of health, and material circumstances. 

Due to the specificity of the analyzed variables, two non-parametric sets of statistics were used: Spearman’s rank correlation coefficient and the Mann–Whitney U test. A value of *p* < 0.05 was adopted as the level of significance. All data attained in the research was analyzed statistically using the IBM SPSS Statistics for Windows, Version 25.0 (IBM Corporation, Armonk, NY, USA).

## 3. Results

The median score of the prestige declared for the paramedic profession, on a scale of 1 (very low) to 5 (very high), was 4.49, which placed the profession in fourth place in the ranking, following the profession of firefighter (4.67), doctor (4.66), and university professor (4.51). If we consider only the highest level of prestige that respondents gave to individual professions, expressed by the grade of 5, then the paramedic comes in third place, after firefighter and doctor (Figure 1).

The assessment of respect for paramedics, among other medical professions, placed them in third place, following doctors and midwives (Figure 2). Herein, 63.7% declared a very high respect for the representatives of this profession, expressed with the highest score on the scale of 5, whereas doctors gathered 74% and midwives gathered 63.8%, i.e., only 0.1% more than paramedics, of such scores. This confirms the stable position and high valuation of the paramedic profession, both on the scale of prestige and in the area of its emotional components, as measured by the assessment of respect. 

According to our own research, every second Pole (51%) believed that the salary of a paramedic is high (23%) or very high (18%), while every third respondent assessed them at an average level, and only 15% thought they were low or very low (9.2% were unable to assess the paramedics’ earnings).

The assessment of the prestige of the paramedic profession is correlated with the sociodemographic characteristics of the respondents. People living in larger cities and who were better at assessing their income, tended to express greater respect for the profession of paramedic. At the same time, people who estimated the paramedic’s income as higher, expressed a higher level of recognition for the profession and a greater respect for its representatives (Table 2). 

## 4. Discussion

Our own, nationwide, representative research is, so far, the only one on such a scale in which the position of various medical professions was analyzed, including that of paramedics. In these studies, Poles granted a very high social prestige to the profession of paramedic, locating it in third place, both on the scale of regard and on the scale of respect. In the aforementioned American studies [43,44,45], paramedics also ranked above average, which confirms their relatively high valuation. In addition, our own representative research demonstrated that paramedics are among the highest-ranking professions. In the few available analyses of the social perceptions of the profession from other countries (although not always representative and applying different methodologies, which makes unambiguous comparisons impossible), we found confirmation that it is respected and trusted all over the world, and patients highly value the competence of paramedics [46,47,48,49]. In Australian studies, members of local communities perceived paramedics as professionals who were well-educated, with a great sense of responsibility for the health and life entrusted to them; thus, they expect a high level of assistance from the profession [50]. In both European and American studies, patients rated their contact with paramedics either well or very well [47], and few comments concerned the possibility of improving interpersonal communication [48]. Although, in most studies, only a narrow category of ‘satisfaction’ with contact with the paramedic was analyzed, as noted by Perry et al., the high ratings of this satisfaction are a consequence of the attitudes of paramedics helping the patient regain a sense of confidence and order in the chaos and uncertainty in which they found themselves in the relationship with their traumatic health event [51].

In 2019, paramedic teams accomplished almost 3.1 million ambulance/air ambulance tasks in Poland, 72.2% of which were calls to patients’ homes. However, paramedics provided medical assistance to only 81 out of 1000 inhabitants of Poland [52]. This means that only a slight percentage of respondents had direct contact with the paramedic profession. We may thus assume that most of the information about the daily work of paramedics comes from the media, which has thus significantly shaped their image. Here, social respect for the profession could have been raised by media coverage of spectacular rescue operations, presenting the paramedics’ activities in terms of heroism and special missions. The media shows them as responsible, heroic, empathic, resourceful, and hardworking people who, in the face of a threat to human health and life, both during the day and at night, have to deal with difficult professional situations. 

The picture of the daily work of paramedics, however, can be quite different than the one shown in idealized media broadcasts: difficult working conditions, stress and professional burnout, claimant patients, the physical and verbal aggression the paramedics often face, low sense of control [53,54,55], and, at the same time, unsatisfactory earnings— in reality, all these situations accompany the profession of paramedic.

It is also worth stressing that the profession of firefighter occupies first place in the rankings of most prestigious professions in Poland [56,57], which could indicate that it is the very rescue function that is particularly valued by Poles. At the same time, the prestige of the profession of paramedic is ranked higher by dwellers of larger cities, where the ‘visibility’ of paramedics is surely higher than in towns or in rural areas.

Still, it seems that, despite the granting of high prestige, many objective conditions are not taken into account in determining the social image of paramedics’ work. The amount of earnings is usually an important component of the prestige assessment, and studies have demonstrated that a high rating of prestige and respect for the profession of paramedic is clearly associated with a belief in the high level of their income. People who indicated that the profession had high earnings gave paramedics a status level of greater prestige. However, it turns out that respondents overestimate the earnings of paramedics. More than half said that they were either high or very high but, according to public information, a paramedic with higher education employed in Poland as part of the Medical Rescue System can count on a wage of only 60% of the national average [58]. 

The low salaries of paramedics, combined with overwork, fatigue, and the stress accompanying their daily work cause more and more cases of resignation from work within the Medical Rescue Teams or A&E departments, and the subsequent taking up of work not related to the learned profession. 

The situation of the paramedic can be treated as an example of the imbalance between the prestige of a profession and the income derived from its exercise. Higher education, growing qualifications, and scope of independence, as well as high social recognition associated with low pay, is a classic example of ‘status inconsistency’. As Meyer and Hammond claim, status inconsistency—conflict between several evaluated attributes of a given individual—is seen as occurring with respect to common social rules that define appropriate combinations of status characteristics [41].

What is particularly noticeable in the case of the profession of paramedic is the inconsistency between the individual attributes of the status. The expected resources are relatively high (higher education, state examination), the cultural evaluation of the status is high, as indicated by results of research that situate the profession of paramedic high on the prestige scale, and the market/economic value of paramedic’s efforts is huge, as its subjects are human health and life. Unfortunately, both the rewards connected with this status are low, and the strength, power, and social influence locate this profession at a low level.

Poles highly value the profession of paramedic, granting it great respect, but its social position, as expressed by objective measures (earnings, lifestyle, structural possibilities, social power), is significantly lower. 

The shortage of subject literature and empirical comparative material means that this article is only exploratory in its character. It reveals the prestige of the profession and outlines the problem of dissonance between this, as expressed by social respect for the profession, and its socio-economic status. We included selected medical and non-medical professions in our comparison. In subsequent editions of the study, the issue of the social situation of the profession should be specified and the criteria of prestige and its socio-cultural determinants indicated. It would certainly be interesting to determine the level of Poles’ knowledge of the role of a paramedic, its competences, scope of professional activities, and responsibilities and then tie that level of respondents’ awareness with the evaluation of prestige of the profession. It would also be interesting to make international comparisons regarding the social place of the paramedic profession, its prestige, and social perception, especially during and after the current pandemic. 

## 5. Conclusions

The paramedic, despite being a relatively young profession, is characterized by high social prestige, locating it at the forefront of the medical and other examined professions. The prestige of the profession of paramedic is ranked higher by dwellers of larger cities, where the ‘visibility’ of paramedics is surely higher than in towns or in rural areas. Social perceptions of the profession and its prestige determine a patient’s trust and attitudes to the paramedic as a member of the interdisciplinary team working for the health of the patient, particularly in emergency situations.

## Figures and Tables

**Figure 1 ijerph-18-01506-f001:**
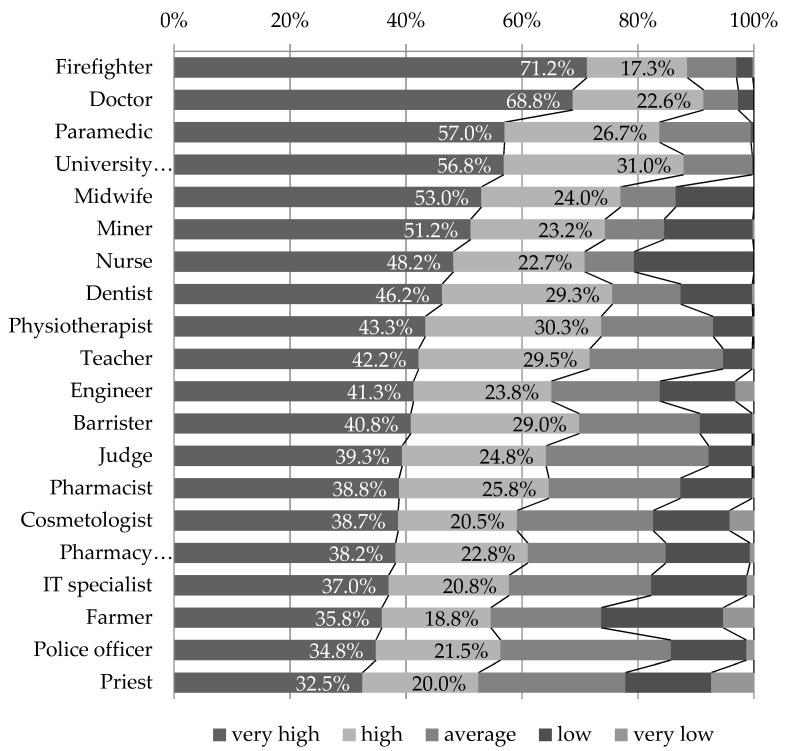
Prestige of professions, as expressed in the ‘respect for profession’ grade.

**Figure 2 ijerph-18-01506-f002:**
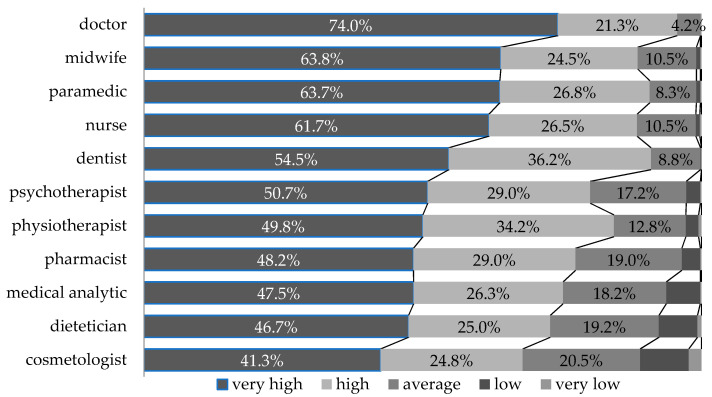
Respect for selected medical professions in the opinion of Poles.

**Table 1 ijerph-18-01506-t001:** Characteristics of the research group.

Feature	Demographic Variables	N	%
**Gender**	Women	306	51.0%
	Men	294	49.0%
**Age**	≤29	106	17.7%
	30–39	126	21.0%
	40–49	102	17.0%
	50–59	98	16.3%
	60+	168	28.0%
**Education**	Primary	13	2.2%
	Basic vocational	116	19.3%
	Secondary	212	35.3%
	Bachelor’s degree	131	21.8%
	Master’s degree	128	21.3%
**Place of residence**	Village	33	5.5%
	Town with a population of up to 20,000	60	10.0%
	City with a population of between 20,000 and 100,000	228	38.0%
	City with a population of between 100,000 and 500,000	225	37.5%
	City with a population of 500,000 or more	54	9.0%
**Self-assessment of health**	Very good	144	24.0%
	Good	372	62.0%
	Average	76	12.7%
	Poor	8	1.3%
**Assessment of material circumstances**	Very good	68	11.3%
	Good	359	59.8%
	Average	166	27.7%
	Poor	7	1.2%

**Table 2 ijerph-18-01506-t002:** Relationship between prestige of the paramedic profession with selected variables—Spearman correlation coefficient.

	Median	Respect for the Paramedic	Respect for the Paramedic among Medical Profession
**Respect for paramedics**	4.49		
**Respect for paramedics among the medical profession**	4.60	0.596 **	
**Age**	44.58	0.028	0.012
**Education**	3.38	0.051	−0.035
**Place of residence**	3.41	0.095 *	0.070
**Self-assessment of health**	4.12	−0.003	0.030
**Assessment of material conditions**	3.80	0.223 **	0.232 **
**Assessment of paramedics’ income**	3.47	0.248 **	0.217 **

** *p* < 0.01; * *p* < 0.05.

## Data Availability

The data presented in this study are available on request from the corresponding author.

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
