# Peer review of "Social Prestige of the Paramedic Profession"

_ijerph, 2021, doi:10.3390/ijerph18041506_

Round 1

Reviewer 1 Report

Thank you for addressing the issues raised in the first review. This is a much improved paper and now gives a real insight into why respect and status are viewed as being important. This is very much helped by the additional narrative relating to professionalization. The paper also starts to hint at patient benefits in terms of patient attitude and on to the potential for improving trust between patient and paramedic.

The point of prestige and respect being used interchangeably adds clarity. I wonder if it would be useful to note this in the abstract too.

I still have some reservations regarding sampling and the potential exclusion of those who do not have telephone access

One minor point - it would be helpful to avoid the abbreviations e.g and write in full in the narrative. This does not change the meaning but it make the flow of the narrative less disruptive 

In table 1 - Education; I do not know what 'gymnasial' means

Author Response

Dear Reviewer,

We would like to thank you for careful and thorough reading of our manuscript and for the thoughtful comments and constructive suggestions, which helped to improve the quality of our manuscript. We appreciate the positive feedback from you.

We did our best to apply all suggestions and comments into manuscript.

The point of prestige and respect being used interchangeably adds clarity. I wonder if it would be useful to note this in the abstract too.

We did it – the explanation is in the abstract too. 

One minor point - it would be helpful to avoid the abbreviations e.g and write in full in the narrative. This does not change the meaning but it make the flow of the narrative less disruptive 

Corrected in the text.

In table 1 - Education; I do not know what 'gymnasial' means

We left just ‘primary’ as 'gymnasial' was an error. Thank you for the note.  

I still have some reservations regarding sampling and the potential exclusion of those who do not have telephone access

We understand the Reviewer’s point of view, but as already mentioned in the manuscript the CATI technique of telephone survey was only applied to supplement missing data, and accounted for 16% of the entire sample. Additionally, according to Office of Electronic Communications Report,  more than nine out of ten Poles use a mobile phone. We assume, the potential exclusion of those who do not have telephone access is minimal.

Thank you very much for helping us to improve the quality of the manuscript.

Regards,

Anita Majchrowska

Reviewer 2 Report

This paper has improved substantially following revisions. I now think it is ready for acceptance following some very minor edits as follows:

Abstract - The opening line, please delete 'Both in Poland, and in Europe', it is not needed. Final line of abstract, 'locating it at the forefront', that is not quite correct, as the paper does (importantly) mention that the public status of the paramedic is high, but the actual status (in terms of pay, conditions, autonomy) is somewhat lower. This sentence should be revised to show that key finding.

Page 2 - Section about Freidson is good, but it could be revised to be clearer. The phrasing is a bit clumsy. Please consider revising to improve clarity.

Page 2 - 'In the 1979 Polish study' should be 'In a 1979 Polish study'

Page 4 - 'The data comes from' should be 'Data are derived from'

Page 8 - 'honored' probably not the right word. Consider revising

Page 8 - 'to him' - Please revise to use gender-neutral language.

Page 8 - 'Civilization development' - not really the right phrase. In fact, this whole sentence is redundant, maybe just cut it and start the paragraph with the next sentence?

Page 8 - 'departures' should be 'calls'

Page 9 - Paramedics in NHS are not 'entitled to early retirement'. But they very often are granted early retirement on medical grounds (due to physical injury, stress, etc).

Page 9 - first sentence of section 5 contains typographical errors. 

Lastly, let me say this is an interesting paper, and I hope to see it published. Good luck with it.

Author Response

Dear Reviewer,

We would like to thank for careful and thorough reading of our manuscript and for the thoughtful comments and constructive suggestions, which helped to improve the quality of our manuscript. We appreciate the positive feedback from the you. 

We did our best to apply all suggestions and comments into manuscript.

This paper has improved substantially following revisions. I now think it is ready for acceptance following some very minor edits as follows:

Abstract - The opening line, please delete 'Both in Poland, and in Europe', it is not needed. Final line of abstract, 'locating it at the forefront', that is not quite correct, as the paper does (importantly) mention that the public status of the paramedic is high, but the actual status (in terms of pay, conditions, autonomy) is somewhat lower. This sentence should be revised to show that key finding.

Page 2 - Section about Freidson is good, but it could be revised to be clearer. The phrasing is a bit clumsy. Please consider revising to improve clarity.

Page 2 - 'In the 1979 Polish study' should be 'In a 1979 Polish study'

Page 4 - 'The data comes from' should be 'Data are derived from'

Page 8 - 'honored' probably not the right word. Consider revising

Page 8 - 'to him' - Please revise to use gender-neutral language.

Page 8 - 'Civilization development' - not really the right phrase. In fact, this whole sentence is redundant, maybe just cut it and start the paragraph with the next sentence?

Page 8 - 'departures' should be 'calls'

Page 9 - Paramedics in NHS are not 'entitled to early retirement'. But they very often are granted early retirement on medical grounds (due to physical injury, stress, etc).

Page 9 - first sentence of section 5 contains typographical errors. 

We definitely agree with all the comments and suggestions of the reviewer. All of them are corrected in the manuscript.

Lastly, let me say this is an interesting paper, and I hope to see it published. Good luck with it.

Thank very much for helping us to improve the quality of the manuscript.

Regards,

Anita Majchrowska

This manuscript is a resubmission of an earlier submission. The following is a list of the peer review reports and author responses from that submission.

Round 1

Reviewer 1 Report

Lines 31-34; You seem to have taken a very narrow view of the role of the paramedic, specifically emergency or life threatening situations Later you refer to technical expertise both from a procedure perspective and the drugs formulary. This is really not reflective of the role internationally and certainly in the UK wher only about 10% of calls. I would suggest that patients and families are interested in the technical competence of the paramedic, but are also interested in care, compassion and other personal qualities and values. I would anticipate that these factors also have an impact on their perception of the the value and status of paramedics

Line 34; 'vitals' is an informal term and may not be understood by a wider readership.

Line 35; Professionalism and profesionalization is about so much more than technical expertise.

Line 41; You may wish to use UK rather than England as the statement would equally apply to Scotland, Wales and Northern Ireland

Line 43; You make reference here to prestige and position, presumably status, but have not really said why this is important. This is a key thrust of your paper and it would be important to understand early on why the status of paramedics is materially important. The following narrative tells us what prestige is but not why it is important.

Line 58: were paramedics analysed or was the prestige of the role analysed?

Lines 55-63; it is not clear if the same scale was used, making it difficult to make comparisons?

Line 67; the reader will not be able to make a judgement as the nature of the study and data collection is not known.

Table 1; It would be helpful to see the national demographic alongside the sample demographic. I have concerns about sampling, and whether the sample is representative of the population. There are strong claims of this but it is not clear that there is a strong enough foundation for this claim. This may be a methodological flaw or limitations in the description of the sampling. I am unable to make a judgement and neither will the readership. I am also sceptical that the variables identified to attest to the subjects being representative are sufficient. There is no rationale as to why these variables were selected.

Lines 94-95; It seems those without a telephone would be excluded from the study. How are subjects identified if several people are associated with a single telephone number.

Line 108; your data collection tool refers to respect. How does this align with status and prestige? It is not clear how you have made the link or association between respect and prestige. The value subjects place on paramedics is not really addressed and could be an important factor. There seems to be a use of proxy measures which is not made clear, discussed and evaluated.

Had the subject had contact with paramedics or other professional groups which may have influenced their rating?

Statistical analysis

Your data is ordinal and so non-paramedic test can be used, but in line 126 you talk of average, by which I assume is mean average. There is little value in the mean when applied to ordinal data, for example a mean of 4.4 the respect for paramedics. It is impossible to  determine what the mid point between large and very high respect is in real terms. I would suggest that a data collection tool gathering ratio data may be of more use. If rank order was important then you could simply ask subjects to put each profession in order and then test for internal consistency.

Results

You have used the term fireman and firefighter. You should be consistent, but firefighter is probable the more acceptable of the 2 terms.

You have interchanged respect for prestige - without an explanation, this is somewhat confusing

Table 2; Again, the use of median in the context of ordinal data is unhelpful. It may be OK for age as this is likely to be ratio data

This section makes the first reference to a correlational design. It need to be considered in the analysis section

You have introduced both prestige and respect, but it is not clear how this relates to the questions identified in the data collection tool

Discussion and conclusion

It is difficult to evaluate this as the narrative is based on what I believe to be a flawed design.

Reviewer 2 Report

This is an interesting paper on a timely topic. The survey of Polish citizens’ opinions of paramedics is useful and fairly detailed. I think the paper has potential to make a useful contribution to knowledge. However, it also has several shortcomings, all of which have to do with the interpretation of the survey and the overall purpose of the paper; both of which could be clearer.

The paper is about public opinion and a clinical profession’s status. But it does not define its terms and concepts sociologically. It mentions ‘status’, ‘prestige’, ‘the social structure’, ‘professions’, but does not link any of this into an established literature. There is no mention of the sociology of professions literature (Abbott The System of Professions or Freidson Profession of Medicine, etc ). Use of literature of this kind might help the authors to orient and define their concepts and, therefore, to better set out their paper’s broader analysis and argument. I think this is the central issue that the paper needs to look at. What is ‘a profession’ and how will we know that paramedics are now part of one? There is no mention of the classic ‘traits’ of a profession, such as professional associations, journals, codes of practice, etc (see the early chapters of Leicht & Fennell Professional Work, or more recently Muzio et al Professional Occupations and Organizations). The most problematic area left out of the paper is that of education. This is surely a fundamental issue? Back when the EMT / paramedic model was developed, training was in first aid, manual handling, emergency driving, etc, lasting a few weeks and trained on the job. Now, the trend is towards new paramedics being registered with a professional license and regulated, with entry onto the register often now requiring a degree in paramedic science. Laypersons generally don’t know this and this issue won’t be picked up in a study using this kind of design.

The paper acknowledges the role of media in framing laypersons’ (partial) views of the paramedic profession, but more could be done to develop the analysis. One of the reasons that paramedics hold a fascination with the public is that the public believes their work is dangerous, high-stakes, exciting, dramatic, etc. But this perspective captures only a small part of what they do. High-acuity trauma calls are at the root of the traditional emergency paramedic model and are central to old style  training, but trauma calls are increasingly less a part of paramedics main activity. Paramedics are increasingly moving into unplanned primary care, and into ‘grey area’ type calls (concern for welfare / psychiatric calls). Trauma calls are becoming less common relative to the rest of their work. The paper might want to speculate about how the public perception of paramedics might change as their practice expands into non-emergency areas. This could conceivably either move them into a higher bracket as they become more clinically valued (like doctors) or a lower one (as their role become less recognizably one of ‘hero’ and ‘rescuer’). A brief discussion of this could perhaps go into the section about limitations of the study and areas for future analysis.

There should also be a clearer discussion of what a survey of this kind can’t tell us – the public’s views are interesting, but often ill-informed. It does raise the question of what we can (and can’t) learn from a study design of this kind. It is likely that layperson’s views of any profession are very partial, and are gleaned from general knowledge, guesswork, impressions from media, etc. So what exactly is the paper trying to argue? The survey tells us that the general public in Poland is very respectful of paramedics and that they are surprised about how little they earn. But the same could be said for nurses. Why is this an issue? What, at least, are the policy implications for paramedics, public health and pre-hospital medicine?

The paper cites some important studies in the paramedic journals, but does not mention any of the (growing) work on paramedics from social scientists, e.g. the work of Josh Seim, Morten Kyed, Michael Corman, Leo McCann, Edward Granter, Maree Boyle, Paresh Wankhade, all from studies of paramedics across USA, Canada, UK, and Australia. This work (across various papers and books) provides extensive detail about the social status of the paramedic profession, and its movement from an occupation rooted in first aid and transportation to its current guise as a profession with growing clinical scope. I would recommend that the authors read and cite some of this sociological material in order to enhance and develop the paper’s broader impact. The paper’s findings could be strengthened if they are related more closely to what we already know from other sociological accounts of paramedics and their role in society.

Minor editing suggestions:

‘fireman’ should be ‘firefighter’

‘Mass media’ is a very old-fashioned terminology and should be revised.

There are a few references to the paramedic as ‘rescuer’ which also fits the hero / trauma narrative but is not really accurate as to their clinical role. The authors could consider revising this.

Overall, I enjoyed reading this paper and I thin there is much to recommend it. However, I think, at present the paper does not provide enough detail as to the broader meaning of their survey findings. It also needs to better define its sociological terms and to embed the work into a broader discussion of professional ‘traits’, and the existing sociological literature about paramedics and their work. I wish the authors the best of luck with the paper.